# Bayesian Hierarchical Modelling of Historical Data of the South African Coal Mining Industry for Compliance Testing

**DOI:** 10.3390/ijerph19084442

**Published:** 2022-04-07

**Authors:** Felix Made, Ngianga-Bakwin Kandala, Derk Brouwer

**Affiliations:** 1School of Public Health, Faculty of Health Sciences, University of the Witwatersrand, Johannesburg 2193, South Africa; ngianga-bakwin.kandala@wits.ac.za (N.-B.K.); derk.brouwer@wits.ac.za (D.B.); 2Global Biostatistics and Programming, Pharmaceutical Product Development, Thermo Fisher Scientific, Woodmead, Johannesburg 2191, South Africa; 3Warwick Medical School, University of Warwick, Coventry CV4 7AL, UK; 4Département de la Santé Communautaire, Institut Supérieur des Techniques Médicales de Kinshasa, Kinshasa XI, Mont Ngafula, Kinshasa B.P. 774, Congo

**Keywords:** non-informative prior, informative prior, 95th percentile, lognormal distributions, exposure category

## Abstract

Bayesian hierarchical framework for exposure data compliance testing is highly recommended in occupational hygiene. However, it has not been used for coal dust exposure compliance testing in South Africa (SA). The Bayesian analysis incorporates prior information, which increases solid decision making regarding risk management. This study compared the posterior 95th percentile (P95) of the Bayesian non-informative and informative prior from historical data relative to the occupational exposure limit (OEL) and exposure categories, and the South African Mining Industry Code of Practice (SAMI CoP) approach. A total of nine homogenous exposure groups (HEGs) with a combined 243 coal mine workers’ coal dust exposure data were included in this study. Bayesian framework with Markov chain Monte Carlo (MCMC) simulation to draw a full P95 posterior distribution relative to the OEL was used to investigate compliance. We obtained prior information from historical data and employed non-informative prior distribution to generate the posterior findings. The findings were compared to the SAMI CoP. The SAMI CoP 90th percentile (P90) indicated that one HEG was compliant (below the OEL), while none of the HEGs in the Bayesian methods were compliant. The analysis using non-informative prior indicated a higher variability of exposure than the informative prior according to the posterior GSD. The median P95 from the non-informative prior were slightly lower with wider 95% credible intervals (CrI) than the informative prior. All the HEGs in both Bayesian approaches were in exposure category four (poorly controlled), with the posterior probabilities slightly lower in the non-informative uniform prior distribution. All the methods mainly indicated non-compliance from the HEGs. The non-informative prior, however, showed a possible potential of allocating HEGs to a lower exposure category, but with high uncertainty compared to the informative prior distribution from historical data. Bayesian statistics with informative prior derived from historical data should be highly encouraged in coal dust overexposure assessments in South Africa for correct decision making.

## 1. Introduction

South Africa (SA) is one of the largest producers of coal in the world, with an estimated 86,000 workers according to the Mineral Council of South Africa, 2019 [1]. During coal mining, coal dust is generated, and when inhaled, it can cause coal mine lung disease (CMDLD) [2,3]. To lower the risk of coal dust overexposure and potentially prevent life-threatening CMDLD, the occupational exposure limit for respirable coal dust in SA is set at 2 mg/m^3^ [1,4]. As part of a continuous process of monitoring overexposure and compliance, the South African Mining Industry Code of Practice (SAMI CoP) stipulates that the identification of homogenous exposure groups (HEGs) is an important proxy for the assessment of personal exposures [5]. HEGs are defined as a group of employees who have similar exposure, such that a sample can be drawn from them for predicting the exposure of all remaining workers [5,6]. In the SAMI CoP, HEGs are constituted by a stepwise process [5,6]; in step one, the mine is subdivided into ventilation districts based on areas with common intake and return air. In the next step (step 2), the area is divided into activity areas (as found in coal mines). The personal exposures in each activity area are then compared to the OEL, which is the eight-hour time-weighted average (TWA8h) coal dust concentration to which almost all workers may be repeatedly exposed without any adverse health effects whatsoever. Each HEG is assigned to exposure classification categories, which are associated with their distance to the OEL [4] (Table 1).

The sampling size of each HEG is equal to 5 or 5% of the HEG population, whichever is greater, whereas the sampling frequency is determined by the exposure classification category. The results of each sampling campaign are evaluated independently of the previous data (i.e., historical data) for compliance. This current practice in South Africa shows that HEGs are too heterogeneous concerning exposure levels, resulting in an overestimation or underestimation of exposure for individual workers [7]. Yet, in good practice, previous sampling results can be used to update current data for the categorization of HEGs by using a Bayesian framework to elucidate prior information from the historical data [8]. The framework with broad use of informative prior is highly encouraged in occupational hygiene because it accommodates historical data into the empirical measurement of monitoring exposure data for accurate exposure grouping [8,9,10,11,12]. 

For compliance testing, the SAMI CoP approach assumes that the sample data are normally distributed, and the 90th percentile of exposure data (P90) should be below the OEL. Overexposure according to the European Standardization Committee (CEN) [13] and the British and Dutch guidelines is that the 95th percentile (P95) of the lognormal exposure distribution should be below the OEL [14]. It is important to note that SAMI CoP is based solely on current data, yet the incorporation of historical data in the Bayesian framework could improve the identification of overexposed HEGs as future risk management of exposure is based on the exposure profile of the data at hand. The SAMI CoP approach is based solely on a point estimate of the P90 of the current data and does not consider uncertainty surrounding the estimate. The Bayesian framework uses the credible interval to describe uncertainty surrounding a parameter. For example, the credible interval is interpreted as the probability of an estimate being found between a certain range, given the data [15]. The SAMI CoP approach would mean that similar exposure in different areas needs to be done repeatedly, but in a Bayesian sense, this can be achieved naturally by using a sample result as a prior distribution from common population distribution [16]. The Bayesian inference is exact and easily understood by anyone. For example, the probability that a person is overexposed to coal dust is 95%. Currently, the Bayesian framework has not been applied for exposure to coal dust in the mining industry, and no study has emphasized the use of historical data in determining overexposure in a routine occupational hygiene assessment. Therefore, the first objective of this study was to compare the posterior P95 of the non-informative and informative application of the Bayesian framework and SAMI CoP relative to the OEL. The second objective was to compare the grouping of posterior probabilities of the P95 exposure according to the South African occupational exposure categories between the two Bayesian approaches. The present paper describes the development of an informative prior, taken from the historical coal dust exposure data, and combines this with the present exposure data to achieve posterior distributions in a Bayesian framework. The decision-making on exposure risk management according to SAMI exposure categories is compared to posteriors derived from a non-informative prior and the strengths and limitations are discussed.

## 2. Methods

### 2.1. Study Design and Data Collection

This is a cross-sectional study. Respirable coal dust exposure data were collected periodically from different geographic locations of the mines. The population included in this study were only male mine workers who were working in underground coal mines. The data were collected from workers working in HEGs within each mine.

Aligned with the SAMI CoP approach, a mixed stratified and random selection sampling frame was used, considering that “either 5% of the workers assigned to a HEG, or a minimum of five workers should be selected for a measurement campaign” [5]. The sampling collection and sample analysis methods have been described in a previous paper [7]. Briefly, each selected worker was issued with a size-selective cyclone with a mixed cellulose ester filter, which was attached to a dust sampling. The analysis of the cyclone’s filter was done according to the standard of the National Institute for Occupation Safety and Health (NIOSH) method 0600 [17].

### 2.2. Statistical Analysis

Statistical analysis was carried out in R version 4.1.1(R Core Team, Auckland, New Zealand), using RStan and bayestestR packages [18,19,20]. Consistent and similar coal dust historical data of the HEG was used to update current monitoring data to produce posterior geometric mean, geometric standard deviations, P95, and the posterior probabilities of P95 exposure in each of the exposure categories. For the prior specification, we randomly selected a prior sample size of five from each HEG’s historical data, as recommended from previous studies for occupational exposure assessment [21,22]. From the occupational exposure perspective, the prior sample size should be between 10% to 40% of the current data to get accurate information on the posterior distribution for decision making. Therefore, the sample size of five was used to keep the focus of the posterior distribution on the current data, as the sample size increased. This is important as in Bayesian statistics, the posterior distribution is a compromise between the information from the prior and the current data, but the distribution must be observed from the current data to a good measure as the sample size increases [15]. For the likelihood function, we took all the available current monitoring data. 

#### 2.2.1. Model Specification Using Current Monitoring Data

The model was specified by using the geometric mean (GM), given as exp(*μ*), and geometric standard deviation (GSD), denoted as exp(*σ*), which are the exponents of the mean and standard deviation, after log transforming the data. The likelihood function was presented as below in Equation (1).
(1)$$\overset{n}{\underset{i=1}{\prod }}LN\left(y_{i}|\mu ,\sigma^{2}\right)=\overset{n}{\underset{i=1}{\prod }}\frac{1}{y_{i}\sigma \sqrt{2\pi }}exp\left\{-\frac{1}{2}\frac{{\left(logy_{i}-\mu \right)}^{2}}{\sigma^{2}}\right\}$$
where $y_{i}$ is the log-transformed current monitoring data and *n* is the number of observations of current monitoring data. The OEL exposure categories were added as a random variable in the model directly to produce the posterior probability distribution of the P95 to each of the categories [21,22]. From the OEL exposure categories (Table 1), the highest category was assumed to have P95 > OEL and the lowest was P95 < 10% of the OEL. 

#### 2.2.2. Model Specification Using a Non-Informative Uniform Prior Distribution

In occupational health research, uniform prior distribution is highly encouraged. A current monitoring data vector, $Y=(y_{1},\ldots ,\ldots ,\ldots ,y_{n}$), where *n* is the sample size, with the data $Y\sim Norm\left({\mu \sigma }^{2}\right)$, where *μ* represents the log of the geometric means (GM) and *σ* is the log of the geometric standard deviation (GSD).

Then, $\mu =LnGM\sim Unif\left(a_{\mu }b_{\sigma \mu }\right)$, $\sigma^{2}=LnGSD\sim unif\left(a_{\sigma }b_{\sigma }\right)$ where *a* and *b* are the lower and upper bounds of the prior distribution, respectively. We took inspiration from Gelman 2006 [23], where, for lower bounds, *μ* was indicated as 0 and for *σ* was −1/2 and the upper bounds were set to infinity.

#### 2.2.3. Informative Prior Specification from Historical Data

For informative prior, we assumed that the log-transformed historical data with *n*_0_ observations had sample variance $s_{y0}^{2}=\sum {\left(y_{i0}-{\bar{y}}_{0}\right)}^{2}/\left(n_{0}-1\right)$. If the historical data $y_{i0}∼\mathrm{Norm}\left(\mu ,\sigma^{2}\right),$ then the mean of the historical data ${\bar{y}}_{0}∼\mathrm{Norm}\left(\mu ,\sigma^{2}/n_{0}\right),$ we put *μ* as a random quantity and replace *σ*^2^ with the prior estimate $s_{y0}^{s}$, so that *μ* takes the form as shown below in Equation (2).
(2)$$\mu ∼\mathrm{Norm}\left({\bar{y}}_{0},s_{y0}^{s}/n_{0}\right)$$

The full conditional for μ and $\sigma^{2}$ was based on truncated normal prior distributions and truncated inverse gamma prior distributions, respectively [9,22]. In the truncated prior distributions, we placed bounds on μ using the suggestion of Bayesian decision analysis (BDA) from Hewett et al., 2006 [12], which was 0.005. We used 0.001 in this study to make sure it less likely affected the results, while the upper bound was allowed to vary iteratively. The upper bound was allowed to vary to avoid a prior from being unfairly skewed toward a more favourable result.

For $\sigma^{2}$, the lower bound from BDA was used and the upper bound was let to change iteratively.

To develop prior for $\sigma^{2}$, we started with the expression, $\left(n-1\right)s_{y0}^{2}/\sigma^{2}\sim x_{n_{0-1}}^{2}$, where $x_{n_{0-1}}^{2}$ represents the chi-square distribution with (*n*_0_ − 1) degrees of freedom. We put $\sigma^{2}$ as the random variable, given $s_{y0}^{s}$ from the historical data. Therefore, the variance is given by Equation (3).
(3)$$\sigma^{2}\sim IG\left(\frac{n_{0}-1}{2},\frac{\left(n_{0}-1\right)s_{y0}^{2}}{2}\right)$$

For $n_{0}>1$, where IG (a, b) is an inverse gamma distribution in Equation (4) with parameter *a* and scale parameter *b*

Therefore,
(4)$$a=\left(n_{0}-1\right)/2\;b=2/(s_{y0}^{2}*\left(n_{0}-1\right))$$

Further details on the prior specification for μ and $\sigma^{2}$ and the full conditional distributions are available in Appendix A of Quick, 2017 [22].

#### 2.2.4. Motivation on Bounds Based on P95

We placed the bound on the P95 as P95 < 2 × OEL with the assumption that the correct OEL exposure category for each HEG would be identified. See Appendix A for motivation on bounds based on P95 by Quick, 2017 [22].

Markov chain Monte Carlo (MCMC) algorithms to draw full posterior conditional distribution inform of the Gibbs sampler [24] were implemented. The Gibbs sampler was applied because of its easy computational application. The Gibbs sampler samples from a conditional distribution. For example, if a given parameter has been divided into sub-parameters, the Gibbs sampler works by drawing each sub-parameter conditional on the values of all the others iteratively. The sub-parameter is updated conditional several times on the latest values of all the components of the parameter to produce the marginal posterior distribution. We used 20,000 MCMC number of iterations to draw samples from the posterior.

The posterior convergence diagnostic was carried out using the Gelman–Rubin convergence diagnostic, which compares the between and within-Markov chain variability for the model parameters to confirm whether they are stationary [25]. The between-Markov is the variance of the posterior mean of the samples, while the within-Markov is the mean of the variance in each sample. If the test statistics denoted by R-hat is ≤1.05, then convergence is achieved. The reliability of the posterior quantiles was confirmed using the bulk and tail effective sample size [26]. An effective sample size greater than 100 per chain is considered good. The convergence diagnostic test indicated an R-hat of less than 1.05 and an effective sample size of more than 100 in this study (not shown), implying convergence was achieved.

## 3. Results

Table 1 indicates the exposure classification of SAMI CoP. The highest category is four, which is that P90 exceeds the OEL for exposure to be classified as poorly controlled, and the lowest category is that P90 should be less than 0.1 of the OEL for highly effective control. In Table 2, a summary of the current monitoring data and corresponding historical data from nine HEGs are displayed. HEGs C and G had the highest AM of 2.42 in the current monitoring data compared to the rest of the HEGs. In the past data, HEG A had the highest AM of 2.00. HEGs D and G have the lowest AMs of all the other HEGs in current and past data, respectively. GSD indicated a high variability of exposure with the current monitoring data in HEG B, D, E, F, H, and I (GSD > 3), while in the past historical data, the variability of the exposure was high in HEG B, D, F, G, and I.

Table 3 indicates the SAMI P90, median, and 95% credible intervals of the posterior GM and GSD, and the P95 percentiles for non-informative and informative prior. The SAMI CoP P90 values are much lower than the P95 in the non-informative and informative prior Bayesian approaches. The SAMI approach is the only method that showed that only one HEG (HEG D) had P90 values lower (1.62 mg/m^3^) than the OEL of 2 mg/m^3^. The posterior median GM indicated that all HEGs exposures were below the OEL 2 mg/m^3^. There was high exposure variability in the majority of HEGs, as indicated by GSD greater than 3. Three and four of HEGs under non-informative and informative prior had less exposure variability according to the GSD. The patterns of the medians of the posterior P95 and 95% credible intervals (CrI) from Table 3 are shown in Figure 1. Generally, the median and 95% CrI were similar across all HEGs between the non-informative and informative prior. Five out of nine HEGs in the graph indicated that P95 are lower in the non-informative prior with wider 95% CrI bounds compared to the informative. Overall, there was high uncertainty in the non-informative indicated by the wider 95% CrI (also higher upper bounds) compared to the informative prior distribution. 

The comparison in the grouping of the HEGs’ posterior probabilities of the P95 according to the different OEL categories (see Table 1) is presented in Table 4. In both Bayesian approaches of the prior distribution, HEG D showed the lower posterior probability of the exposure level being in category four, which is poorly controlled compared with the rest of the HEGs. All the HEGs in both prior distributions were in poorly controlled category four with more than 90% and 95% probabilities, respectively. Some of the posterior probabilities of the non-informative prior distribution, although all in category four, were slightly lower than in the informative prior.

## 4. Discussion

We used informative prior from historical data to update current monitoring data on lognormal distribution in the Bayesian framework to produce the posterior geometric mean, geometric standard deviation, and the P95. Similarly, a non-informative prior was used, and SAMI CoP was based on P90. The findings were compared. The posterior probability of the P95 percentile exposures was also grouped according to the SAMI exposure category. The use of the past data is important because decision-making on exposure risk management only based on the current data might be misleading. The weight of the historical data in the analysis is important; Symanski et al. [27] used an equal weight for both current and past data. We decided that weight should be unequal, with a small prior sample size to have less influence on the current data. This is consistent with other studies where small prior sample size was thought to produce inferential benefits when the results were compared to non-informative priors [22,28]. The Bayesian framework is also known to be robust with a small sample size [15], so even in the paucity of data, exposure risk analysis can be conducted with relative confidence.

The application of both approaches to the prior distribution indicated that the posterior estimates of GM were below the OEL. This means that the level of coal dust risk control was similar in the past and the present. However, risk mitigation and decision making regarding exposure control should not be based on the central distribution (mean/median) of the data, but on P95, which constitute at least 95% of the underlying distribution. The posterior GSDs were also quite similar across the two Bayesian prior distributions, however, those of the non-informative distributions tended to be somewhat higher. The GSD indicated high variability in the informative prior distribution. The comparison of the SA SAMI using the P90 for compliance and Bayesian prior methods showed that P90 was lower (with one HEG exposure below the OEL) compared to P95. This is consistent with our previous study, where the SAMI approach tended to underestimate overexposure risk [4]. The Bayesian approaches considered the uncertainty of overexposure not just based on a point estimate as to the SAMI CoP. All the HEGs in the Bayesian approach indicated their P95 were very high and above the OEL. The distribution of the median and P95 was similar to the non-informative and informative prior distribution (Figure 1). The majority of HEGs in the non-informative prior indicated that P95 were a bit lower and had wider 95% CrIs, indicating a high uncertainty compared with the informative prior derived from the historical data. This underscores the importance of the use of historical data in coal mining occupational exposure assessment. Decisions on overexposure risk can be made with greater confidence when historical data are brought together to update the current data, as it is only natural that they are part of the current data. 

We then compared the posterior probabilities of grouping the P95 in each exposure category between the Bayesian approach from non-informative and informative prior (from historical data) distribution (Table 4). In both approaches, the highest probabilities (greater than 95%) of P95 were observed in category four of the exposure, which indicates poorly controlled exposure. None of the HEGs’ posterior P95 was in the lower exposure category. From these results, the use of historical data to update current data in Bayesian statistics for occupational exposure assessment is very important, as non-informative prior tend to assign HEGs’ in a lower category. This affects informed decision making with the regard to overexposure risk mitigation. Assigning HEGs to a lower category is similar to a simulation study that showed that non-informative uniform priors group the P95 probabilities in lower exposure categories [22]. The difference with the informative prior from historical data might be because of the possible use of incorrect prior for certain HEGs, resulting from a lack of adequately repeated measurements and sampling of prior data.

As seen from the above, the uncertainty to inform risk management decisions is not low and high variability of exposure is shown by the non-informative prior distribution compared to the informative prior from the historical data. Although this study showed that non-informative priors tend to locate the posterior probabilities of P95 in a lower category and increase variability, its interpretation must be taken with caution as the probability density function used to specify the prior, usually an infinite integral might yield improper posterior distributions [29]. Therefore, the specification of the non-informative uniform prior must be considered. Our findings could indicate that the decision to regard these HEGs as compliant or non-compliant should also consider the variability of the data. 

The strength of this study is that the Bayesian analysis naturally allows for combining prior information from historical data with current data within a solid decision-making framework [30]. With the robustness of the findings based on even a small sample size, the Bayesian analysis provided inferences that are based conditionally on the data, which makes them exact and easily interpretable. For example, the probability of the posterior P95 being in category four (poorly control group) can be expressed quantitatively [31]. Regarding the limitations of this study, it is important to recall that HEGs used in this study are created by grouping workers based on the common air intake and return air. This might mean that the HEGs can be too heterogeneous because within a HEG there can be several job titles with different exposure variabilities [7]. As demonstrated earlier [7], HEGs tend to have high variability, which also affects their compliance to the OEL and grouping according to exposure category. From the Bayesian perspective, sometimes historical data are unavailable or are not similar and consistent to the current data, and hence it is not possible to conduct an informative prior.

## 5. Conclusions

It is clear from the findings that the use of the Bayesian framework with informative prior can inform concise decision making on occupational exposure risk mitigation in the coal mining industry with great confidence. Bayesian analysis from the non-informative uniform prior distribution tends to put HEGs in lower exposure categories than informative prior distribution derived from historical data. The non-informative prior findings also showed high uncertainty and variability, thus a decision on exposure risk assessments would likely be made with less confidence. This makes overexposure risk likely to be underestimated. We recommend increased use of the Bayesian framework with the use of prior information from historical data in the coal mining occupational exposure assessment. This will improve solid decision-making concerning coal dust overexposure risk and compliance.

## Figures and Tables

**Figure 1 ijerph-19-04442-f001:**
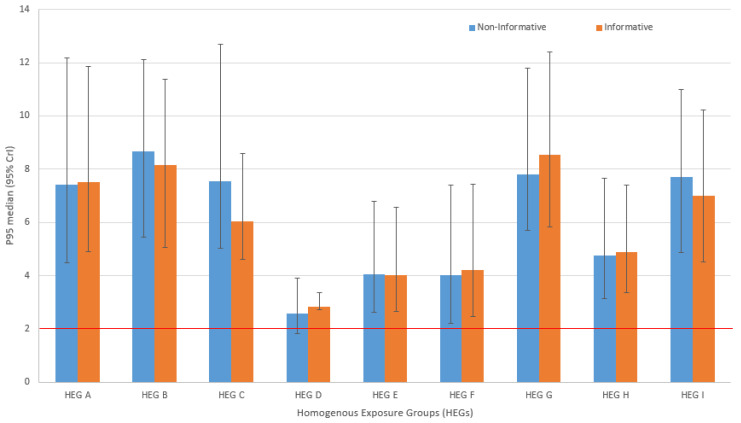
The comparison of the patterns of the posterior median (95% CrIs) of the P95 for non-informative and informative Bayesian framework across HEGs. The red horizontal is the SA OEL of 2 mg/m^3^.

**Table 1 ijerph-19-04442-t001:** The exposure classification categories for classifying the P90 of exposure relative to the OEL and sampling strategy of HEGs in the SAMI CoP.

Category	Description	Statistical Illustration	Exposure Profile	Minimum Frequency
1	Exposures less than 10% of the OEL 10% of the time	P90 < 0.1% OEL	Very highly controlled	No sampling plan for this category. Measurement results that are below 10% of the OEL will be reported under this category
2	Exposures exceed 10% of the OEL and less than 50% of the OEL 10% of the time	P90 ≥ 0.1 OEL < 0.5 OEL	Highly controlled	Sample 5% of employees within a HEG on an annual basis with a minimum of 5 samples per HEG, whichever is greater.
3	Exposures exceed 50% of the OEL and less than OEL 10% of the time	P90 ≥ 0.5 OEL < OEL	Adequately controlled	Sample 5% of employees within a HEG on a 6-monthly basis with a minimum of 5 samples per HEG, whichever is greater
4	Exposures exceed the OEL 10% of the time	P90 ≥ OEL	Poorly controlled	Sample 5% of employees within a HEG on a 3-monthly basis with a minimum of 5 samples per HEG, whichever is greater

**Table 2 ijerph-19-04442-t002:** Summary of coal dust exposure for the current monitoring data and their corresponding historical past data.

Data	Year	N	AM	SD	GM	GSD
Current data						
HEG A	2018	14	2.20	1.96	1.52	2.55
HEG B	2019	21	2.36	1.45	1.58	3.56
HEG C	2018	13	2.42	2.22	1.91	1.96
HEG D	2017	52	0.71	0.66	0.42	3.29
HEG E	2018	35	1.32	1.74	0.62	3.53
HEG F	2018	20	1.24	1.90	0.60	3.80
HEG G	2019	24	2.42	1.70	1.93	2.01
HEG H	2018	38	1.43	1.75	0.74	3.50
HEG I	2019	26	2.04	1.58	1.29	3.67
Past data						
HEG A	2017	20	2.00	1.35	1.51	2.30
HEG B	2018	21	1.93	2.52	0.69	5.96
HEG C	2017	19	1.48	0.95	1.20	2.03
HEG D	2016	53	1.46	1.69	0.76	3.59
HEG E	2017	50	1.18	1.01	0.78	2.78
HEG F	2017	32	0.96	0.82	0.60	3.05
HEG G	2018	40	0.69	0.91	0.29	4.63
HEG H	2017	45	0.90	0.82	0.62	2.44
HEG I	2018	39	1.02	0.91	0.59	3.27

N: sample size; AM: athematic mean; SD: standard deviation; GM: geometric mean; GSD: geometric standard deviation.

**Table 3 ijerph-19-04442-t003:** The median (95% credible interval (CrI)) of the posterior GM, GSD, and the P95 percentiles and the SAMI P90.

		Non-Informative			Informative		
HEG	SAMI P90	GM	GSD	P95	GM	GSD	P95
		Median (95% CrI)	Median (95% CrI)	Median (95% CrI)	Median (95% CrI)	Median (95% CrI)	Median (95% CrI)
HEG A	4.12	1.47 (0.86, 2.33)	2.67 (2.29, 3.25)	7.42 (4.48, 12.17)	1.56 (1.05, 2.28)	2.59 (2.28, 3.06)	7.50 (4.91, 11.85)
HEG B	4.02	1.40 (0.82, 2.13)	3.01 (2.66, 3.48)	8.67 (5.44, 12.11)	1.24 (0.72, 1.91)	3.12 (2.77, 3.59)	8.14 (5.07, 11.36)
HEG C	3.74	1.89 (1.24, 2.78)	2.33 (2.02, 2.86)	7.55 (5.04, 12.70)	1.63 (1.26, 2.08)	2.21 (1.97, 2.64)	6.03 (4.60, 8.60)
HEG D	1.62	0.42 (0.30, 0.59)	3.01 (2.73, 3.40)	2.58 (1.83, 3.92)	0.46 (0.34, 0.65)	2.99 (2.73, 3.36)	2.83 (2.73, 3.36)
HEG E	3.27	0.62 (0.41, 0.96)	3.11 (2.76, 3.63)	4.04 (2.61, 6.78)	0.62 (0.41, 0.93)	3.11 (2.78, 3.60)	4.02 (2.65, 6.57)
HEG F	2.46	0.58 (0.31, 1.02)	3.23 (2.76, 3.93)	4.01 (2.22, 7.41)	0.63 (0.38, 1.03)	3.16 (2.75, 3.76)	4.20 (2.47, 7.43)
HEG G	4.24	1.93 (1.43, 2.60)	2.34 (2.10, 2.72)	7.79 (5.70, 11.79)	1.66 (1.09, 2.39)	2.70 (2.41, 3.12)	8.53 (5.83, 12.40)
HEG H	4.02	0.74 (0.49, 1.11)	3.09 (2.76, 3.57)	4.76 (3.14, 7.66)	0.79 (0.57, 1.10)	3.01 (2.71, 3.44)	4.88 (3.36, 7.40)
HEG I	4.06	1.19 (0.73, 1.80)	3.09 (2.74, 3.54)	7.72(4.88, 10.98)	1.07 (0.66, 1.62)	3.12 (2.77, 3.59)	6.99 (4.50, 10.21)

P95: 95th percentile; CrI: credible Interval; OEL is ≤2 mg/m^3^.

**Table 4 ijerph-19-04442-t004:** The estimated exposure category probabilities of the non-informative and informative Bayesian frameworks for the posterior 95th percentile.

HEG	Non-Informative					Informative				
	P95	Category 1	Category 2	Category 3	Category 4	P95	Category 1	Category 2	Category 3	Category 4
HEG A	7.42	0	0	0	100%	7.50	0	00	00	100%
HEG B	8.67	0	0	0	100%	8.14	0	0	0	100%
HEG C	7.55	0			100%	6.03	0	0	0	100%
HEG D	2.58	0	0.01%	7.73%	92.26%	2.83	0	0	2.26%	97.74%
HEG E	4.04	0	0	0.10%	99.90%	4.02	0	0	0.06%	99.94%
HEG F	4.01	0	0.02%	1.07%	98.91%	4.20	0	0	0.27%	99.70%
HEG G	7.79	0	0	0	100%	8.53	0	0	0	100%
HEG H	4.76	0	0	0.01%	99.99%	4.88	0	0	0	100%
HEG I	7.72	0	0	0	100%	6.99	0	0	0	100%

## Data Availability

The data presented in this study are available on request from the corresponding author.

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
