# Peer review of "Bayesian Hierarchical Modelling of Historical Data of the South African Coal Mining Industry for Compliance Testing"

_ijerph, 2022, doi:10.3390/ijerph19084442_

Round 1

Reviewer 1 Report

As the authors say, Bayesian hierarchical framework for exposure data compliance testing is highly recommended in occupational hygiene. In "Bayesian Hierarchical Modeling of Historical Data of the South African Mining Industry for Compliance Testing" they compared the posterior 95th percentile (P95) of the Bayesian non-informative and informative prior from historical data relative to the occupational limit exposure (OEL) and exposure categories, and the South African Mining Industry Code of Practice (SAMI CoP) approach.

This is, of course, a public health problem, so this work is entirely relevant to IJERPH.

The study begins with an introduction and description of the problem, accompanied by a review of the relevant literature, which provides adequate information and support for the development of the study. What is missing here, at the end of the Introduction, is a commented description of the organization of the article, as is usual. This situation has to be corrected.

In section 2.Methods, the methodology for collecting the sample is described in great detail; also the quantitative tools used are rigorously described, and there is nothing to prevent your choice.

Section 3 presents the results that appear cleanly and clearly.
In section 4 the results are discussed in great detail and clarity.

Section 5 presents the conclusions and corresponding recommendations.

But, references to the strengths and weaknesses of the research and possible leads for future research on this subject are lacking. Some attention has to be paid to this situation.

The writing is simple and smooth, in good English. It will certainly be an easy read.

Publish after the minor revisions suggested.

Reviewer 2 Report

The objective of this work is to apply Bayesian modeling to a 5% sample of the South African Mining Industry. To do so, they collect data on the air breathed by mine workers. 

They should clarify in both the abstract and the introduction how they are going to achieve their objective. 

Why do they choose 5% of the population?

Why do they assume that the samples are normally distributed and not with other types of distributions?

Why the 90% percentile?

It is not clear what type of data is used in the introduction and summary section.

They should list the equations

The conclusions could incorporate the main contribution of the paper and the advantages of applying Bayesian modeling to this particular case. 

Reviewer 3 Report

Review Report for Bayesian Hierarchical Modelling of Historical Data of the South African Mining Industry for Compliance Testing in IJERPH.

The goal of this paper is to compare the posterior 95th percentile (P95) of the Bayesian non-informative and informative prior from historical data relative to the occupational exposure limit (OEL) and exposure categories, and the South African Mining Industry Code of Practice (SAMI CoP) approach. The authors’ topic sounds interesting. But I have some suggestions that will make this work publishable in IJERPH. My comments are below:
Major Comments:

Introduction
The introduction presents the paper’s objectives and other important issues. However, the paper’s contributions are missing. Why is this research important? Why should readers care? After reading it, perhaps the use of a Bayesian method could be one of their contributions?  I recommend authors refer to Gelman et al. (2013)(https://doi.org/10.1201/9780429258411) and this important paper that uses a hierarchical Bayesian method from which they can benefit a lot in this regard(https://www.mdpi.com/2073-4395/10/3/376). The reference above could be helpful. For example, why a Bayesian approach rather than the commonly used classical econometrics?

Methods
1) In section 2.2: Authors say: In R with the Bayesian package… This is incorrect. What R Bayesian package did you use? There are over a million R packages out there and so please mention the package used, and cite the it and its developers here as well as the R software.

2) In section 2.2: Authors claim the sample size of 5 is to ensure the prior does not overpower the prior. This is incorrect. The sample size is so small and it depends on which of the data or prior overpowers the other by the type of the prior used. It is a compromise between the two. So authors should restate this statement because a small sample size of 5 has little chance of overpowering the prior and especially that they use informative priors. The paper above mentions something on Bayesian methods’ benefits for sample sizes. Authors can benefit by supporting the use of Bayesian methods even for small sample size as theirs as indicated in the above paper.

3) Authors suggest the prior for sigma squared was specified by using methods from conventional statistics. Authors should explain why? What are conventional statistics? This ambiguous. Bayesian statistics is not conventional statistics and it is only Bayesian statistics that focuses on putting a prior on a parameter. In short, rewrite that statement.

4) What values did a and b take in the Inverse Gamma prior? Please explain.

5) Authors mention they used MCMC for their estimations. They should explain briefly explain how the Gibbs sampler works. There is also Metropolis-Hastings and should explain why chose Gibbs sampler?

6) Authors say they used R-hat and effective sample size to check reliability of their MCMC samples. Authors should explain what these things are and what made them sure their chains converged. This is important for Bayesian analysis. Again, authors should refer to the paper I recommended above.

Minor Comments
Discussion:
The first two paragraphs under this section: discuss the results and their implications.
Implications are missing. Authors are mentioning the results again.

Conclusion: Authors say: However, the use of historical data must be treated with caution to avoid overly informative prior. What limitations does this paper have? I think the use of informative priors is one of them? Since they are discouraging its use? In short, explain any limitations.

Round 2

Reviewer 3 Report

Authors considered the comments.